# Real-World Evidence of Treatment-Free Remission Strategies and Outcomes in Chronic Myeloid Leukemia

**DOI:** 10.3390/cancers17132148

**Published:** 2025-06-26

**Authors:** Garrett Bourne, Kendall Diebold, Greg Bascug, Joshua Knapp, Manuel Espinoza-Gutarra, Pankit Vachhani, Kimo Bachiashvili, Sravanti Rangaraju, Razan Mohty, Ravi Bhatia, Omer Jamy

**Affiliations:** 1Department of Medicine, University of Alabama at Birmingham, Birmingham, AL 35294, USA; gabourne@uabmc.edu; 2Division of Hematology and Oncology, Department of Medicine, University of Alabama at Birmingham, 1720 2nd Avenue S, NP2540W, Birmingham, AL 35294, USA; kdiebold@uabmc.edu (K.D.); ogbascug@uabmc.edu (G.B.); jpknapp@uabmc.edu (J.K.); mgutarra@uabmc.edu (M.E.-G.); pvachhani@uabmc.edu (P.V.); kbachiashvili@uabmc.edu (K.B.); srangaraju@uabmc.edu (S.R.); rmohty@uabmc.edu (R.M.); rbhatia@uabmc.edu (R.B.)

**Keywords:** CML, treatment-free remission, TKI strategies

## Abstract

One of the recent goals for the treatment of chronic myeloid leukemia (CML) is for patients to be able to discontinue their treatment to attain treatment-free remission (TFR). In our study, we report on real-world outcomes of patients with CML entering TFR at our institution. Furthermore, we describe three strategies to manage tyrosine kinase inhibitors (TKIs) prior to discontinuation; standard-dose cessation, standard-dose tapering, and upfront dose reduction. Our results demonstrate that TFR can be safely pursued by any of these strategy in a real-world setting. Among the three TKI management strategies, safety outcomes were comparable, with no instances of disease progression or CML-related mortality. Our study highlights the relative safety of pursuing TFR via different TKI treatment strategies in a real-world setting.

## 1. Introduction

Chronic myeloid leukemia (CML) is a myeloproliferative condition with uncontrolled proliferation of granulocytes in the blood and bone marrow. It is characterized by the reciprocal translocation of chromosomes 9 and 22, resulting in an abnormally short chromosome 22 (Philadelphia chromosome (t (9; 22) (q34; q11.2)) [1], creating *BCR::ABL1*, a constitutively active tyrosine kinase oncoprotein that results in uncontrolled hematopoietic cell proliferation. Historically, CML has had a poor prognosis with an 8-year overall survival rate of <20% [2,3,4]. In 2001, the United States Food and Drug Administration (FDA) approved the first therapeutic tyrosine kinase inhibitor (TKI) designed specifically to inhibit *BCR::ABL1*, which significantly improved outcomes [5]. Since 2001, several different generations of TKIs have been approved, and, as a result, CML in its chronic phase is now considered by many to have been virtually cured [3,4,6].

However, TKIs come with adverse effects (AEs). Depending on the specific TKI, patients can experience a variety of AEs ranging from fatigue, edema, rashes, gastrointestinal toxicities, and muscle aches to more serious AEs such as pleural/pericardial effusions, pancreatitis, and peripheral arterial occlusive disease [1,7,8]. Furthermore, TKIs add to healthcare costs, and many patients face financial burdens as a result of their CML treatment. Given these issues, questions have arisen regarding the efficacy of taking patients with sustained deep molecular remission (DMR) off their TKIs to see if they could maintain a treatment-free remission (TFR).

The very first prospective trial designed to assess TFR efficacy was the Stop Imatinib Trial (STIM1), which evaluated 100 patients who had achieved undetectable *BCR::ABL1* levels for >2 years. The results were promising, with 38% of patients remaining in TFR at 60 months, no patients progressing or dying CML-related deaths, and 96% of patients recovering a major molecular response (MMR) if and when imatinib was restarted [9]. Since then, additional prospective studies have attempted to identify the most effective methods to obtain and then sustain TFR by evaluating various TKI treatment methods. To date, the data from these various studies is heterogenous, without an obviously more reliable or superior method. Current guidelines from both the National Comprehensive Cancer Network (NCCN) and European LeukemiaNet (ELN) provide recommendations on when it is reasonable to attempt TFR, but questions remain regarding which TKI treatment methods will make the most patients eligible for TFR and allow them to maintain a sustained TFR.

We conducted this study to not only address the ongoing knowledge gap regarding preferred TKI treatment methods prior to TFR but also provide real-world outcomes for patients attempting TFR at a large National Cancer Institute (NCI)-designated comprehensive cancer center in the southeast United States.

## 2. Methods

We performed retrospective analysis comparing the outcomes of patients with chronic-phase CML (CP-CML) entering TFR either via abrupt cessation of a TKI at a standard dose, TKI dose tapering prior to cessation, or upfront TKI dose reduction followed by abrupt cessation. Data were obtained from patients seen at the University of Alabama at Birmingham (UAB) between June 2001 and December 2024 after obtaining institutional review board (IRB) approval. Standard TKI doses were defined as imatinib 400 mg daily, dasatanib 100 mg daily, and nilotinib 300 mg twice a day. Variables captured included age, gender, race, European Treatment and Outcome Study for CML (EUTOS) long-term survival score (ELTS), and Sokal risk scores, as well as the TKI and dose at which it was used. Patients with accelerated-phase or blast-phase CML were excluded.

Outcomes of interest included median duration of TFR, median time to loss of TFR, median time to regain MMR after TFR loss, and the rates of patients who remained in active TFR. TFR loss was defined as the loss of MMR [*BCR::ABL1* international standard (IS) >0.1%]. Data on rates of disease progression, CML-related death, and recurrence of MMR/molecular response (MR4)/MR4.5 after TKI reinitiation were also captured.

We calculated summary statistics, including the median and range for continuous variables and frequencies and percentages for categorical variables. The Student *t* test was employed to compare continuous variables and the X^2^ test was utilized to compare categorical variables. Survival outcomes were estimated using the Kaplan–Meier method. All analyses were conducted using IBM SPSS Statistics 30.0.

## 3. Results

We identified 233 patients with chronic-phase CML. Of those, 46 patients (19.7%) were eligible to attempt TFR either according to the NCCN (version 2.2024) or ELN 2020 criteria. Of the 46 eligible patients, 44 patients met the ELN criteria for TFR and 2 patients met the NCCN criteria but not the ELN criteria. Of the eligible patients, 29 attempted TFR, whereas 17 patients did not attempt TFR. Another 10 patients attempted TFR despite not meeting the NCCN or ELN criteria (Figure 1).

The baseline characteristics of the overall cohort and the TFR cohort are shown in Table 1. Unfortunately, data required to calculate Sokal and ELTS risk scores were missing for the majority of patients. In the TFR cohort, there was an even distribution of the final TKIs utilized—imatinib (33%), dasatinib (39%), and nilotinib (27%).

Of the 39 patients that attempted TFR, 21 entered TFR via abrupt cessation of a TKI at a standard dose, 11 entered via TKI dose tapering prior to cessation, and 7 received an upfront TKI dose reduction, often for ongoing medical comorbidities, followed by abrupt cessation. The baseline characteristics of the three different TKI management strategies prior to TFR are seen in Table 2. For patients undergoing abrupt cessation of a standard-dose TKI, there was increased usage of imatinib as the final TKI (48%) relative to dasatinib (33%) and nilotinib (19%). Of patients undergoing TKI dose tapering, the final TKI utilized by most patients was either dasatinib (45%) or nilotinib (45%), with the rest utilizing imatinib (10%). Lastly, of patients entering TFR following upfront TKI dose reduction, most patients were on dasatinib (71%), followed by imatinib (29%).

As seen in Table 3, a total of 56 CP-CML patients were identified either to have attempted and/or to be eligible for TFR. Of those patients, 39 attempted TFR and 17 were eligible for TFR but did not attempt it, with the most common reason being patient preference. Regarding the 39 patients who attempted TFR, only 74% (*n* = 29) of them met formal eligibility criteria set out by the NCCN and ELN guidelines. Of the ten ineligible patients who attempted TFR, one patient elected to stop therapy while pregnant, one patient with *BCR::ABL1*, detected at diagnosis by cytogenetics and FISH but undetectable by q-RT-PCR, elected to trial TFR with FISH monitoring, two patients had multiple severe comorbidities and adverse effects with several TKIs so elected to attempt TFR, one patient stopped TKI after starting an aggressive chemotherapy regimen for an independent malignancy, three patients elected to attempt TFR after being lost to follow-up without medication for extended periods of time while in DMR, and two patients did not have enough information available for us to determine their eligibility based on TFR criteria.

For the entire TFR cohort, the median time from diagnosis to the TFR attempt was 83 months (8–257). The median duration spent on a TKI prior to TFR and the mediation duration of DMR prior to TFR were 87 months (23–220) and 58 months (0–153), respectively. The vast majority of patients achieved MR4.5 (84%) and had only been on one TKI (82%) prior to attempting TFR.

As noted in Table 4, for all the patients in the TFR cohort, the median TFR duration was 14.6 months. Of the patients who attempted TFR, 63% of them actively remained in TFR, with a median follow-up time of 21 months (3–81) (Figure 2). For those who lost TFR, this occurred at a median time of 5.8 months (2–51). TFR was lost by 55% of patients within the first 6 months, 16% of patients between 6 months and a year, 8% of patients between 1 and 2 years, and 21% of patients after 2 years. Of the patients who restarted on a TKI, all were restarted on their prior TKI and 100% of them regained MMR within a median time of 3 months (1–7). There were zero instances of disease progression or CML-related death. For patients with a loss of TFR after 2 years, 60% were treated with imatinib and 40% with dasatinib, without any other distinguishable characteristics.

Of the three different TKI cessation strategies, the majority (*n* = 21) were treated with a standard-dose TKI followed by abrupt cessation. Of the patients who received a dose de-escalation of their TKI prior to cessation (*n* = 11), this almost exclusively occurred as result of treatment intolerance at standard doses. However, there were a few rare instances (*n* = 2) where the dose was de-escalated to confirm the lack of disease recurrence at a tapered dose prior to fully discontinuing therapy in line with the patient’s preference. As previously mentioned, the few patients (*n* = 7) who were treated with dose-reduced TKIs upfront were typically treated in such a way due to notable medical comorbidities. Between the three different TKI management strategies, safety appears to be the same, with zero instances of disease progression or CML-related death. Furthermore, all patients who lost TFR successfully regained MMR after restarting TKI therapy. Regarding efficacy, it should be noted that of the patients who underwent abrupt cessation of standard-dose TKIs, 61% remained in TFR at a median of 26 months. This was compared to dose de-escalation prior to tapering and upfront dose reduction, where 59% of patients remained in TFR at median durations of 13 and 16 months, respectively (Figure 3). The only notable difference in outcomes for the patients eligible and attempting TFR, when compared to those ineligible and attempting TFR, was the time to loss of TFR (5 m for eligible patients, range 2 m–51 m vs. 2.5 m for ineligible patients, range 1 m–10 m).

The incidence of withdrawal syndrome after TKI discontinuation was 35% in the entire cohort. It was 38%, 31%, and 33% for the abrupt cessation of a standard-dose TKI, dose de-escalation prior to tapering the dose of a TKI, and upfront dose reduction of a TKI, respectively. The median time to onset was 2 months (range 1–4 months). All cases were managed symptomatically without the need for TKI reintroduction.

## 4. Discussion

TKI therapy has profoundly improved the lives of countless CML patients, albeit often at the cost of bothersome AEs and financial burdens. As such, the goal of further advances in CML treatment has moved beyond achieving sustained DMR towards achieving sustained TFR. In pursuing this goal, it is important to elucidate TKI management strategies that will increase the rates of sustained TFR without compromising patient safety.

Our study highlights the relative safety of TFR, with zero incidents of disease progression or CML-related death among the patients who attempted TFR and all patients regaining MMR upon TKI resumption. Further, it identified approximately 63% of patients remaining in TFR at 21 months. For those that lost MMR, we noted that this loss occurred within the first six months for most patients, though upwards of 20% of patients were noted to have lost MMR after two years of TFR. Regarding preferred TKI management prior to TFR, our data indicates that all methods were similarly effective with equivalent safety outcomes.

Our findings regarding TFR safety and duration are comparable with current reports in the literature. The vast majority of prospective trials indicate that no patients who have trialed TFR have died a CML-related death, with disease progression rates of 0–1% and MMR recovery rates of 86–100% [10,11,12,13,14,15,16,17,18,19]. Data from prospective trials on TFR duration are more variable given the lack of consistent endpoints between studies. A few trials have assessed the percentage of patients remaining in TFR at two years, while others have assessed the same percentage at five years, and others have utilized endpoints within this time range. Overall, these studies indicate the percentage of patients remaining in TFR between two and five years to range from 38 to 72%, which is consistent with our findings [12,20,21,22,23,24,25].

When it comes to patients who lose MMR while attempting TFR, our findings vary slightly from the current literature. Two meta-analyses assessing the safety of attempting TFR indicated that the majority of molecular relapses occurred within the first six months, at rates of 80% and 82%, respectively [26,27]. In both studies, it was rare for patients to relapse after two years of TFR. Our findings were consistent with the current literature regarding the majority of patients relapsing within the first six months (55%). However, we also noted a significant number of patients relapsing after two years (21%). The current NCCN and ELN guidelines suggest performing *BCR::ABL1* tests monthly during the first six months, every two months during months seven to twelve, and every three months indefinitely thereafter [7,28]. While current guidelines do recommend indefinite assessments for *BCR::ABL1* recurrence, our findings indicate that providers should remain vigilant in their assessment for disease recurrence even after the two-year TFR mark.

Relative to other aspects of TFR, the current literature on preferred TKI management prior to attempting TFR remains sparse. Most prospective trials have employed abrupt cessation of standard-dose TKIs prior to attempting TFR and have not assessed the possible benefits of TKI dose tapering or upfront TKI dose reduction prior to attempting TFR. Notwithstanding this, both the DESTINY and DANTE studies have indicated that TKI dose tapering is safe, effective, and well-tolerated by patients, which could potentially lead to increased rates of TFR-eligible patients [10,29]. In the DESTINY study, a 50% dose de-escalation of either imatinib, nilotinib, or dasatinib for 12 months, once MMR or MR4 was attained, resulted in feasible TFR outcomes. Although our study demonstrated the feasibility of this approach in patients with deeper remission prior to attempting TFR and doses lower than a 50% reduction, the success of such an approach needs to be prospectively validated, especially in patients with MMR prior to TKI discontinuation [30]. In the DANTE trial, the dose reduction of nilotinib was associated with favorable TFR outcomes in patients with MR4 or deeper remission at the time of trial entry [31]. Our cohort for dose reduction included 45% of patients taking nilotinib and demonstrated somewhat similar outcomes, with the acknowledgement of limited patient numbers. One difference in our cohort was the inclusion of patients on a nilotinib dose of <300 mg per day. Similarly, a 2023 study conducted at MD Anderson Cancer Center indicated that upfront dasatinib reduction is also safe, effective, and well-tolerated and may also lead to increased rates of TFR-eligible patients [32]. Of note, neither study directly assessed the TFR outcomes of patients who had trialed TKI dose tapering or upfront TKI dose reduction. Our results suggest that all three TKI management approaches were similarly effective, with approximately 60% of patients remaining in TFR for each strategy. However, it should be noted that the average follow-up was almost twice as long for the standard-dose abrupt cessation strategy than the other two TKI management approaches. These findings warrant further prospective evaluation.

Our study is not without limitations. Most notable is the retrospective nature of our analysis, which creates potential for various biases to influence the outcomes. We acknowledge the unavailability of risk stratification (Sokal/ELTS) for the majority of the patients in our cohort. This was mainly due to missing data on splenomegaly at the time of diagnosis, as most patients were diagnosed and initially treated at outside facilities and it was challenging to obtain this information. These scoring models could have provided further insights into the outcomes of TFR, and future real-world studies incorporating them would be beneficial. Furthermore, our study was conducted at a single institution with a limited sample size of TFR patients, which limited our ability to generate enough statistical power to identify significant differences between the treatment groups. Lastly, we included patients who attempted TFR despite being ineligible for TKI discontinuation as this is reflective of real-world practice, where patients may discontinue therapy due to various socioeconomic or medical circumstances, as discussed in the results section. We feel it is important to report on such cases to provide insights to colleagues who may experience similar clinical scenarios. We do acknowledge that including such patients could potentially influence the results of the study, which we tried to mitigate by comparing eligible and ineligible patients, although the limited sample size in each arm prohibited a detailed analysis. Nonetheless, this real-world data offers valuable insights into practices in cancer centers in the Deep South, where many patients face healthcare disparities, and should be regarded as an additional resource alongside clinical trial findings.

## 5. Conclusions

In conclusion, our findings emphasize the relative safety of attempting TFR for eligible patients in a real-world setting outside of a clinical trial. Although most patients who underwent molecular relapse did so within the first six months after their TFR attempt, it is possible that more patients than previously expected will relapse >2 y into their remission, emphasizing the need for an indefinite close follow-up for these patients. Lastly, we provided early data suggesting that both the safety and efficacy of various TKI treatment strategies are similar. This suggests that there is not an obviously superior TKI treatment strategy and the decision on which to utilize will likely be centered on each patient’s unique circumstance.

## Figures and Tables

**Figure 1 cancers-17-02148-f001:**
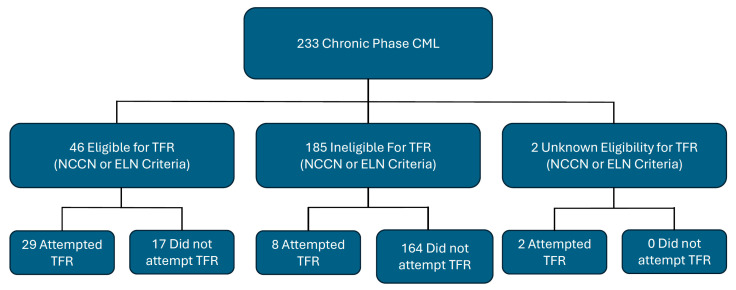
Patient population.

**Figure 2 cancers-17-02148-f002:**
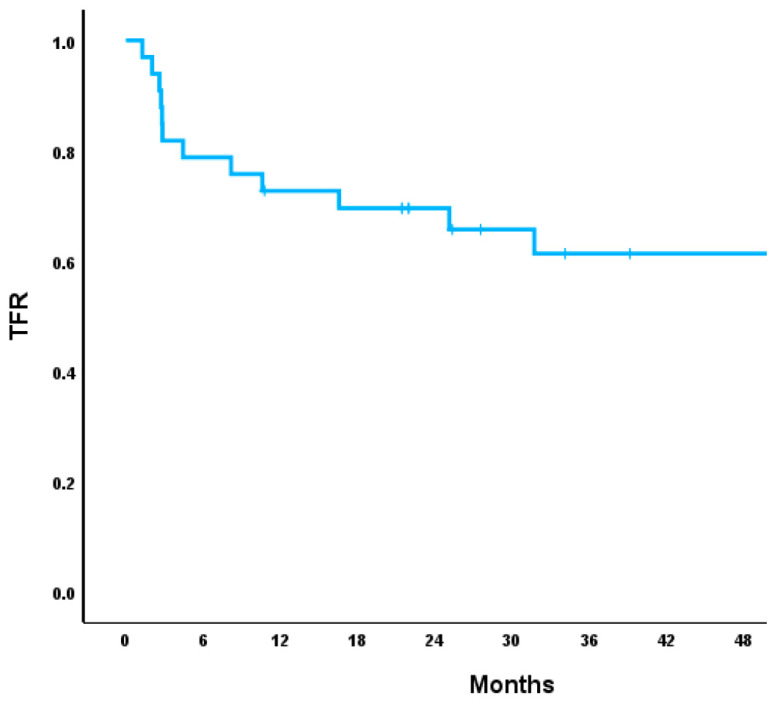
Treatment-free remission.

**Figure 3 cancers-17-02148-f003:**
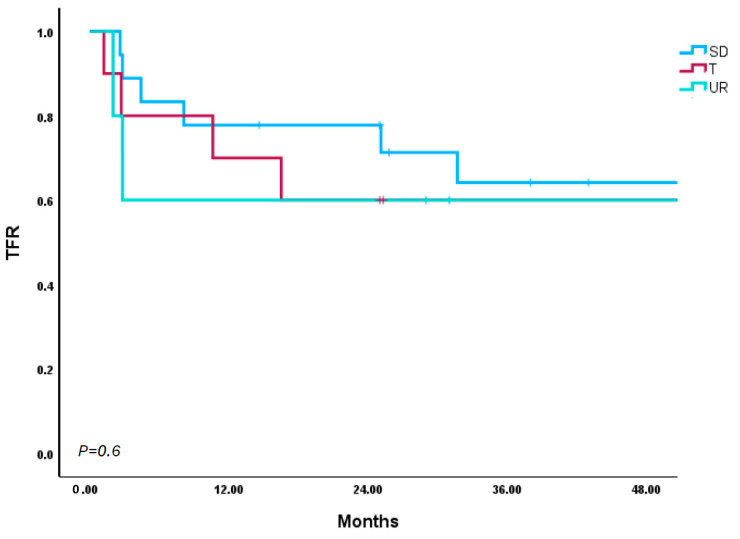
TFR outcomes by TKI management strategies.SD: standard-dose abrupt stop; T: standard-dose taper; UR: upfront dose reduction.

**Table 1 cancers-17-02148-t001:** Baseline characteristics of the overall cohort and the TFR cohort.

Baseline Characteristics	TFR Cohort (*N* = 39)		Full Cohort (*N* = 233)	
**Age in Years (Median)**	47	Range: 20 y–65 y	54	Range: 20 y–72 y
**Gender (%)**				
Female	69		63	
**Race (%)**				
Caucasian	76		72	
African American	20		26	
Asian	4		2	
**Vital Status (%)**				
Alive	89		81	
Deceased	11		11	
Lost to Follow-up	0		8	

TFR: treatment-free remission.

**Table 2 cancers-17-02148-t002:** Baseline characteristics of the three different TKI management strategies prior to TFR.

	Standard-Dose Abrupt Stop (*N* = 21)		Standard-Dose Tapering (*N* = 11)		Upfront Dose Reduction (*N* = 7)	
**Age in Years (Median)**	47	Range: 20–65 y	58	Range: 26–61 y	46	Range: 28–61 y
**Gender (%)**						
Female	65		73		71	
**Race (%)**						
Caucasian			73		71	
African	65		9		29	
American	20					
Asian	10					
**TKI prior to TFR (%)**						
Imatinib	48		10		29	
Dasatinib	33		45		71	
Nilotinib	19		45			
**CML Related Death (%)**						
No	95		91		100	
Lost to Follow-up	5		9			

**Table 3 cancers-17-02148-t003:** Characteristics of patients receiving and/or eligible for TFR.

Total CML-CP Patients Receiving and/or Eligible for TFR (*N* = 56)	
**Number Eligible (NCCN and/or ELN)**	*N* = 46
NCCN Criteria (%)	100 (*N* = 46)
ELN Criteria (%)	96 (*N* = 44)
Unknown (%)	4 (*N* = 2)
**TFR Eligible and Not Attempted (%)**	30 (*N* = 17)
**TFR Attempted**	70 (*N* = 39)
Eligible and Attempted (%)	74 (*N* = 29)
Ineligible ant Attempted (%)	21 (*N* = 8)
Unknown Eligibility and Attempted (%)	5 (*N* = 2)
**Time from Diagnosis to TFR Attempt (Months)**	Median = 83 (8–257)
**Total Duration on TKI Prior to TFR (Months)**	Median = 87 (23–220)
**Duration of Deep Response Prior to TFR (Months)**	Median = 58 (0–153)
**MMR Before TFR (%)**	
MR4	16
MR4.5	84
**Number of TKIs before TFR (%)**	
1	82
2	13
3	5

TFR: treatment-free remission; NCCN: National Comprehensive Cancer Network; ELN: European LeukemiaNet; MMR: major molecular response.

**Table 4 cancers-17-02148-t004:** Outcomes of TFR for the entire cohort and by TKI management strategy.

Prior Therapy	Median TFR Duration (m)	Median Time to Lose TFR (m)	TFR Lost 0–6 m	TFR Lost 6–12 m	TFR Lost 1–2 y	TFR Lost >2 y	MMR Reobtained	MR4.5 Reobtained	Median Time to Reobtain MMR (m)	Disease Progression/CML-Related Death
**Initial TFR for All Participants (*n* = 39)**	14.6 (2.0–78.0)	5.9 (2.0–50.7)	55.00%	16.00%	8.00%	21.00%	100.00%	100.00%	3.1 (0.6–8.4)	0.00%
**Standard-Dose Abrupt Cessation (*n* = 21)**	25.7 (2.9–78.0)	8.1 (2.6–34.3)	50.00%	16.67%	0.00%	33.33%	100.00%	100.00%	3.7 (0.8–8.4)	0.00%
Imatinib (*n* = 10)	30.3 (3.2–78.0)	25.1 (2.6–31.7)	40%	0.00%	0.00%	60%	100.00%	100.00%	5.1 (0.8–6.6)	0.00%
Dasatinib (*n* = 7)	9.8 (2.9–45.6)	7.4 (4.3–13.6)	100.00%	0.00%	0.00%	0.00%	100.00%	100.00%	3.40 (2.9–5.1)	0.00%
Nilotinib (*n* = 4)	9.7 (3.0–39.1)	5.5 (2.8–8.2)	50.00%	50.00%	0.00%	0.00%	100.00%	100.00%	3.0 (2.1–4.9)	0.00%
**Standard-Dose Taper (*n* = 11)**	12.9 (2.7–61.6)	13.6 (2.7–50.7)	25.00%	25.00%	25.00%	25.00%	100.00%	100.00%	1.2 (0.6–2.8)	0.00%
Imatinib (*n* = 1)	11.3 (8.9–25.9)	N/A	N/A	N/A	N/A	N/A	N/A	N/A	N/A	0.00%
Dasatinib (*n* = 5)	25.6 (11.2–54.0)	10.6 (8.5–18.5)	N/A	40%	20%	40%	100.00%	100.00%	1.2 (0.8–4.1)	0.00%
Nilotinib (*n* = 5)	13.8 (2.73–61.6)	9.7 (2.7–16.7)	60.00%	0.00%	40.00%	0.00%	100.00%	100.00%	1.7 (0.6–2.8)	0.00%
**Upfront Dose Reduction (*n* = 7)**	7.3 (2.0–51.6)	3.4 (2.0–5.7)	100%	0.00%	0.00%	0.00%	100.00%	100.00%	4.3 (2.5–7.5)	0.00%
Imatinib (*n* = 2)	4.0 (2.8–5.1)	2.83 (2.5–3.3)	100%	N/A	N/A	N/A	100.00%	100.00%	4.7 (4.3–5.1)	0.00%
Dasatinib (*n* = 5)	19.0 (2.0–51.6)	5.11 (4.5–20.2)	100%	0.00%	0.00%	0.00%	100.00%	100.00%	2.2 (1.5–6.9)	0.00%

TFR: treatment-free remission; MMR: major molecular response; MR4.5: molecular response 4.5.

## Data Availability

The data that support the findings of this study are available upon request from the corresponding author.

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
