# Peer review of "Real-World Evidence of Treatment-Free Remission Strategies and Outcomes in Chronic Myeloid Leukemia"

_cancers, 2025, doi:10.3390/cancers17132148_

Round 1

Reviewer 1 Report

Comments and Suggestions for Authors

The authors retrospectively analyzed outcomes of TFR in patients with CML Although analysis of three fashions for TKI discontinuation; that is, abrupt cessation, dose tapering, and upfront dose reduction is just a little interesting, this article does not provide new information for readers regarding TFR in CML.

Major comments:

  1. The number of analyzable/ eligible patients (33 patients) in this study is too small to draw definite conclusions. Especially, the number of patients who received an upfront TKI dose reduction prior to the cessation was only 4, which may prevent from appropriate statistical analysis.
  2. In relation to above comment, the authors should describe the number of analyzable/ eligible patients in the Abstract.
  3. This manuscript lacks Simple Summary required in this journal.
  4. The results of withdrawal syndromes of TKI such as arthritis in the three fashions for TKI discontinuation is important in the clinical practice.
  5. The preparation of Tables and Figures is very poor.

Figures 1 and 2: The title for respective figures should be under the figures. Figure legends are need, especially for Figure 2.

Tables 1 to 4 lacks legends. Please add them including definition for abbreviations.

The data such as 46.97 should be as 47.0 in all cases, especially in Table 4.

The lines for data are out of place in all Tables.

Minor comments:

  1. For the comprehension of readers, please first write a full-term, then use abbreviation; MMR (line 25), NCI (line 64), IRB (line 71), EUTOS (line 73), IS (line 78), MR4 (line79), ELTS (96), DMR (line 120, also line 124)).
  2. (t(9;22)(q34;q11))→t(9;22)(q34;q11.2)

Author Response

Reviewer 1

The authors retrospectively analyzed outcomes of TFR in patients with CML Although analysis of three fashions for TKI discontinuation; that is, abrupt cessation, dose tapering, and upfront dose reduction is just a little interesting, this article does not provide new information for readers regarding TFR in CML.

Major comments:

  1. The number of analyzable/ eligible patients (33 patients) in this study is too small to draw definite conclusions. Especially, the number of patients who received an upfront TKI dose reduction prior to the cessation was only 4, which may prevent from appropriate statistical analysis.

We thank the reviewer for their comment. At the time of initial submission, we had included patients till June of 2023.  We have now expanded our database till December 2024 to include additional patients. We now report on 233 patients with CML. We found 6 additional patients who attempted TFR, bringing the total number of TFR to 39 patients (21 in standard dose cessation, 11 in standard dose taper and 7 in upfront dose reduction). The results section, figures and tables have been updated to reflect this throughout the paper. Nonetheless, we still acknowledge that the sample size is small in the limitation section in the discussion on page 8. However, this real-world data offers valuable insights into the practice of cancer centers in the Southeast United States, where many patients face healthcare disparities and should be regarded as an additional resource alongside clinical trial findings.

  1. In relation to above comment, the authors should describe the number of analyzable/ eligible patients in the Abstract.

We thank the reviewer for their comment. We have added the number of patients in the abstract.

  1. This manuscript lacks Simple Summary required in this journal.

We thank the reviewer for their comment. We have added the Simple Summary.

  1. The results of withdrawal syndromes of TKI such as arthritis in the three fashions for TKI discontinuation is important in the clinical practice.

We thank the reviewer for their comment. We have now added that ‘The incidence of withdrawal syndrome after TKI discontinuation was 35% in the entire cohort. It was 38%, 31% and 33% in abrupt cessation of standard dose TKI, dose de-escalation prior to tapering TKI and upfront dose reduction of TKI, respectively. The median time to onset was 2 months (range 1-4 months). All cases were managed symptomatically without the need for TKI reintroduction’ to the results section on page 7.

  1. The preparation of Tables and Figures is very poor.

Figures 1 and 2: The title for respective figures should be under the figures. Figure legends are need, especially for Figure 2.

We thank the reviewer for their comment. We have now updated the 2 figures and added figure 3 as well. Legends have been updated. We defer to the journal format regarding placement of title of figure.

Tables 1 to 4 lacks legends. Please add them including definition for abbreviations.

We thank the reviewer for their comment. We have updated the legends for all tables and included abbreviations as well.

The data such as 46.97 should be as 47.0 in all cases, especially in Table 4.

We thank the reviewer for their comment. We have changed the values in Table 4 to one decimal place.

The lines for data are out of place in all Tables.

 We thank the reviewer for their comment. We have requested the journal to format the lines appropriately.

Minor comments:

  1. For the comprehension of readers, please first write a full-term, then use abbreviation; MMR (line 25), NCI (line 64), IRB (line 71), EUTOS (line 73), IS (line 78), MR4 (line79), ELTS (96), DMR (line 120, also line 124)).

 We thank the reviewer for their comment. We have written the full-term for those abbreviations now.

  1. (t(9;22)(q34;q11))→t(9;22)(q34;q11.2)

Corrected.

Reviewer 2 Report

Comments and Suggestions for Authors

This is an interesting study. However, the terminology related to the disease for the “average reader”, should include that chronic myeloid leukemia is a myeloproliferative condition with uncontrolled proliferation of granulocytes in blood and bone marrow. Furthermore, it should be clearly mentioned that the disease is characterized by reciprocal translocation of chromosomes 9 and 22, resulting in an abnormally short chromosome 22 (Philadelphia chromosome).

Author Response

Reviewer 2

This is an interesting study. However, the terminology related to the disease for the “average reader”, should include that chronic myeloid leukemia is a myeloproliferative condition with uncontrolled proliferation of granulocytes in blood and bone marrow. Furthermore, it should be clearly mentioned that the disease is characterized by reciprocal translocation of chromosomes 9 and 22, resulting in an abnormally short chromosome 22 (Philadelphia chromosome).

We thank the reviewer for their comment. We have now added that ‘Chronic myeloid leukemia (CML) is a myeloproliferative condition with uncontrolled proliferation of granulocytes in the blood and bone marrow. It is characterized by the reciprocal translocation of chromosomes 9 and 22, resulting in an abnormally short chromosome 22 (Philadelphia chromosome, (t(9;22) (q34;q11.2)) creating BCR::ABL1, a constitutively active tyrosine kinase oncoprotein, that results in uncontrolled hematopoietic cell proliferation’ to the introduction section on page 2.

Reviewer 3 Report

Comments and Suggestions for Authors

This manuscript provides meaningful real-world data on treatment-free remission in CML, an area of growing clinical importance. One of the major strengths of the study is its pragmatic approach like capturing outcomes from a single U.S. cancer center outside of a controlled trial setting. This makes the findings highly relevant for routine practice, especially in regions with healthcare access disparities.

The authors did a commendable job comparing different TKI strategies - abrupt cessation, dose tapering, and upfront dose reduction which is rarely explored side-by-side. The safety profile reported is reassuring: no cases of disease progression or CML-related deaths occurred, and all patients who relapsed were able to regain major molecular response upon restarting therapy. That kind of recovery rate strengthens the argument for TFR as a viable goal in well-selected patients.

However, the study is not without limitations. First, the retrospective design and small sample size limit the robustness of the findings. The total number of patients attempting TFR was only 33, and comparisons between the three TKI strategies were further constrained by uneven group sizes and unequal follow-up durations. For instance, patients in the abrupt cessation group were followed for nearly twice as long as those in the tapering or dose reduction groups. This makes it difficult to fairly assess long-term TFR durability across strategies.

Another issue is that around one-third of patients who attempted TFR didn’t meet formal NCCN or ELN eligibility criteria. While this reflects real-world practice, it also introduces potential bias and complicates the interpretation of success rates. It would have been helpful if the authors had stratified outcomes more clearly between eligible and ineligible patients.

Lastly, important prognostic information such as Sokal or ELTS scores was missing for many patients. These scores could have provided additional context to interpret relapse risk and TFR outcomes more accurately.

In summary, this is a thoughtful and well-written study that contributes real-world insights into an evolving treatment paradigm in CML. While the conclusions are generally sound, they should be interpreted with caution due to the study’s retrospective nature and modest cohort size. Further prospective research is warranted to validate these findings and refine patient selection criteria for successful TFR.

Author Response

Reviewer 3

This manuscript provides meaningful real-world data on treatment-free remission in CML, an area of growing clinical importance. One of the major strengths of the study is its pragmatic approach like capturing outcomes from a single U.S. cancer center outside of a controlled trial setting. This makes the findings highly relevant for routine practice, especially in regions with healthcare access disparities.

The authors did a commendable job comparing different TKI strategies - abrupt cessation, dose tapering, and upfront dose reduction which is rarely explored side-by-side. The safety profile reported is reassuring: no cases of disease progression or CML-related deaths occurred, and all patients who relapsed were able to regain major molecular response upon restarting therapy. That kind of recovery rate strengthens the argument for TFR as a viable goal in well-selected patients.

However, the study is not without limitations. First, the retrospective design and small sample size limit the robustness of the findings. The total number of patients attempting TFR was only 33, and comparisons between the three TKI strategies were further constrained by uneven group sizes and unequal follow-up durations. For instance, patients in the abrupt cessation group were followed for nearly twice as long as those in the tapering or dose reduction groups. This makes it difficult to fairly assess long-term TFR durability across strategies.

We thank the reviewer for their comment. At the time of initial submission, we had included patients till June of 2023.  We have now expanded our database till December 2024 to include additional patients. We now report on 233 patients with CML. We found 6 additional patients who attempted TFR, bringing the total number of TFR to 39 patients (21 in standard dose cessation, 11 in standard dose taper and 7 in upfront dose reduction). The results section, figures and tables have been updated to reflect this throughout the paper. Nonetheless, we still acknowledge that the sample size is small in the limitation section in the discussion on page 8. However, this real-world data offers valuable insights into the practice of cancer centers in the Southeast United States, where many patients face healthcare disparities and should be regarded as an additional resource alongside clinical trial findings.

We also acknowledge that ‘Our results suggest that all three TKI management approaches were similarly effective with approximately 60% of patients remaining in TFR in each strategy. However, it should be noted that the average follow-up was almost twice as long for the standard dose abrupt cessation strategy than the other two TKI management approaches. These findings warrant further prospective evaluation.’ This information is added to page 8 in the discussion section.

Another issue is that around one-third of patients who attempted TFR didn’t meet formal NCCN or ELN eligibility criteria. While this reflects real-world practice, it also introduces potential bias and complicates the interpretation of success rates. It would have been helpful if the authors had stratified outcomes more clearly between eligible and ineligible patients.

We thank the reviewer for their comment. We now discuss that ‘Lastly, we included patients who attempted TFR despite being ineligible for TKI discontinuation as this is reflective of real-world practice where patients may discontinue therapy due to various socioeconomic or medical circumstances as discussed in the results section. We feel it is important to report on such cases to provide insight to colleagues who may experience similar clinical scenarios. We do acknowledge that including such patients could potentially influence the results of the study and which we tried to mitigate by comparing eligible and ineligible patients although limited sample size in each arm prohibited a detailed analysis.’ This information is now added to page 9 of the discussion section.

In the results section on page 7 we included ‘The only notable difference in outcomes of patients eligible and attempting TFR, when compared to those who are ineligible and attempting TFR, was time to loss of TFR (5m for eligible patients, range 2m-51m vs. 2.5m for ineligible patients, range 1m-10m).’ Other outcomes between the 2 groups were numerically similar.

Lastly, important prognostic information such as Sokal or ELTS scores was missing for many patients. These scores could have provided additional context to interpret relapse risk and TFR outcomes more accurately.

We thank the reviewer for their comment. We now discuss that ‘We acknowledge the unavailability of risk stratification (Sokal/ELTS) for majority of the patients in our cohort. This was mainly due to missing data on splenomegaly at the time of diagnosis as most patients were diagnosed and initially treated at outside facilities and it was challenging to obtain this information. These scoring models could have provided further insight into the outcomes of TFR and future real-world studies incorporating them would be beneficial’ to page 8 of the discussion section.

In summary, this is a thoughtful and well-written study that contributes real-world insights into an evolving treatment paradigm in CML. While the conclusions are generally sound, they should be interpreted with caution due to the study’s retrospective nature and modest cohort size. Further prospective research is warranted to validate these findings and refine patient selection criteria for successful TFR.

 We thank the reviewer for their comment.

Reviewer 4 Report

Comments and Suggestions for Authors

Thank you for the opportunity to review this manuscript entitled “Real-World Evidence of Treatment-Free Remission Strategies and Outcomes in Chronic Myeloid Leukemia”. The study provides valuable real-world insights into TFR strategies in CML, with clinically relevant findings on safety and cessation approaches. However, major revisions are required to strengthen methodological consistency and clarity.

Major Revisions Required:

  1. The authors need to justify inclusion of ineligible TFR attempts (10/33 patients) and discuss potential bias.
  2. Kindly add Kaplan-Meier curves with log-rank tests for TFR duration comparisons.
  3. The authors need to acknowledge small sample sizes (e.g., *n*=4 for dose reduction) and limit overinterpretation.
  4. Address the impact of unavailable Sokal/ELTS scores on risk stratification in the manuscript.
  5. Contrast findings with DESTINY/DANTE trials and highlight need for prospective validation.

Author Response

Reviewer 4

Thank you for the opportunity to review this manuscript entitled “Real-World Evidence of Treatment-Free Remission Strategies and Outcomes in Chronic Myeloid Leukemia”. The study provides valuable real-world insights into TFR strategies in CML, with clinically relevant findings on safety and cessation approaches. However, major revisions are required to strengthen methodological consistency and clarity.

Major Revisions Required:

  1. The authors need to justify inclusion of ineligible TFR attempts (10/33 patients) and discuss potential bias.

We thank the reviewer for their comment. We now discuss that ‘Lastly, we included patients who attempted TFR despite being ineligible for TKI discontinuation as this is reflective of real-world practice where patients may discontinue therapy due to various socioeconomic or medical circumstances as discussed in the results section. We feel it is important to report on such cases to provide insight to colleagues who may experience similar clinical scenarios. We do acknowledge that including such patients could potentially influence the results of the study and which we tried to mitigate by comparing eligible and ineligible patients although limited sample size in each arm prohibited a detailed analysis.’ This information is now added to page 9 of the discussion section.

In the results section on page 7 we included ‘The only notable difference in outcomes of patients eligible and attempting TFR, when compared to those who are ineligible and attempting TFR, was time to loss of TFR (5m for eligible patients, range 2m-51m vs. 2.5m for ineligible patients, range 1m-10m).’ Other outcomes between the 2 groups were numerically similar.

  1. Kindly add Kaplan-Meier curves with log-rank tests for TFR duration comparisons.

We thank the reviewer for their comment. We have added Figure 3 showing KM curves for TFR outcomes by TKI management strategies.

  1. The authors need to acknowledge small sample sizes (e.g., *n*=4 for dose reduction) and limit overinterpretation.

We thank the reviewer for their comment. At the time of initial submission, we had included patients till June of 2023.  We have now expanded our database till December 2024 to include additional patients. We now report on 233 patients with CML. We found 6 additional patients who attempted TFR, bringing the total number of TFR to 39 patients (21 in standard dose cessation, 11 in standard dose taper and 7 in upfront dose reduction). The results section, figures and tables have been updated to reflect this throughout the paper. Nonetheless, we still acknowledge that the sample size is small in the limitation section in the discussion on page 8. However, this real-world data offers valuable insights into the practice of cancer centers in the Southeast United States, where many patients face healthcare disparities and should be regarded as an additional resource alongside clinical trial findings.

  1. Address the impact of unavailable Sokal/ELTS scores on risk stratification in the manuscript.

We thank the reviewer for their comment. We now discuss that ‘We acknowledge the unavailability of risk stratification (Sokal/ELTS) for majority of the patients in our cohort. This was mainly due to missing data on splenomegaly at the time of diagnosis as most patients were diagnosed and initially treated at outside facilities and it was challenging to obtain this information. These scoring models could have provided further insight into the outcomes of TFR and future real-world studies incorporating them would be beneficial’ to page 8 of the discussion section.

  1. Contrast findings with DESTINY/DANTE trials and highlight need for prospective validation.

We thank the reviewer for their comment. We now discuss that ‘In the DESTINY study, a 50% dose de-escalation of either imatinib, nilotinib or dasatinib for 12 months, once MMR or MR4 was attained, resulted in feasible TFR outcomes. Whereas our study demonstrated the feasibility of this approach in patients with a deeper remission prior to attempting TFR and also at doses lower than 50% reduction, the success of such an approach needs to be prospectively validated, especially in patients in MMR prior to TKI discontinuation. In the DANTE trial, dose-reduction of nilotinib was associated with favorable TFR outcomes in patients in MR4 or deeper remission at the time of trial entry. Our cohort of dose-reduction included 45% patients with nilotinib and demonstrated somewhat similar outcomes with the acknowledgement of limited patient numbers. One difference in our cohort was the inclusion of patients on a nilotinib dose of <300mg per day.’ This information has been added to the discussion section on page 8.

Round 2

Reviewer 4 Report

Comments and Suggestions for Authors

Authors Addressed my comments.